# The Reduction of Uromodulin, Complement Factor H, and Their Interaction Is Associated with Acute Kidney Injury to Chronic Kidney Disease Transition in a Four-Time Cisplatin-Injected Rat Model

**DOI:** 10.3390/ijms24076636

**Published:** 2023-04-02

**Authors:** Zheyu Xing, Kunjing Gong, Nan Hu, Yuqing Chen

**Affiliations:** 1Renal Division, Peking University First Hospital, Beijing 100034, China; zheyu@bjmu.edu.cn (Z.X.);; 2Institute of Nephrology, Peking University, Beijing 100034, China; 3Key Laboratory of Renal Disease, Ministry of Health of China, Beijing 100034, China; 4Key Laboratory of CKD Prevention and Treatment, Ministry of Education of China, Beijing 100034, China

**Keywords:** uromodulin, complement factor H, AKI-to-CKD transition, cisplatin, complement activation

## Abstract

Uromodulin is recognized as a protective factor during AKI-to-CKD progression, but the mechanism remains unclear. We previously reported that uromodulin interacts with complement factor H (CFH) in vitro, and currently aimed to study the expression and interaction evolution of uromodulin and CFH during AKI-to-CKD transition. We successfully established a rat model of AKI-to-CKD transition induced by a four-time cisplatin treatment. The blood levels of BUN, SCR, KIM-1 and NGAL increased significantly during the acute injury phase and exhibited an uptrend in chronic progression. PAS staining showed the nephrotoxic effects of four-time cisplatin injection on renal tubules, and Sirius red highlighted the increasing collagen fiber. Protein and mRNA levels of uromodulin decreased while urine levels increased in acute renal injury on chronic background. An extremely diminished level of uromodulin correlated with severe renal fibrosis. RNA sequencing revealed an upregulation of the alternative pathway in the acute stage. Renal CFH gene expression showed an upward tendency, while blood CFH localized less, decreasing the abundance of CFH in kidney and following sustained C3 deposition. A co-IP assay detected the linkage between uromodulin and CFH. In the model of AKI-to-CKD transition, the levels of uromodulin and CFH decreased, which correlated with kidney dysfunction and fibrosis. The interaction between uromodulin and CFH might participate in AKI-to-CKD transition.

## 1. Introduction

Since the 1970s, cisplatin (CP) has been one of the major chemotherapeutic agents applied to solid tumor treatments [1]. Its combination with other anticancer therapies is also thought to have increasingly high potential to solve drug resistance and tumor recurrence [2]. Cisplatin causes tumor cell death by forming covalent bonds with purine bases and interfering with DNA repair during the cell cycle [2,3]. The urinary system is the major excretory pathway of cisplatin. Excessive accumulation of cisplatin leads to aggressive nephrotoxicity, particularly in cells of the S3 segment of the proximal tubule. The effectiveness of tumor therapy and nephrotoxicity are both dependent on the dosage of cisplatin. As a result, renal injury is the most serious adverse effect restricting the application of cisplatin [4].

The prevalence of CP-induced acute kidney injury (CP-AKI) is approximately 31% to 68% in patients using cisplatin [5]. CP-AKI is characterized by tubular damage and acute renal dysfunction [6]. Severe and recurrent AKI episodes induced by repeated use of cisplatin have been confirmed as the prominent risk factors for developing chronic kidney disease (CKD) [7]. Maladaptive repair after AKI is accompanied by vascular rarefaction, chronic inflammation infiltration, nephron loss and renal fibrosis, accelerating the progression to CKD [8]. A single injection of high-dose cisplatin to rodents is a commonly used CP-AKI model. Studies on this model concentrate on injury mechanisms and potential drug testing; however, its high fatality rate impedes long-term observation of AKI-to-CKD transition [9]. Repeated injections of low-dose CP mirror the chemotherapy routines of cancer patients and allow for research on acute-to-chronic kidney injury transition.

Many proteins are produced or decreased in response to nephrotoxicity. Uromodulin (also known as Tamm–Horsfall protein), is a mucoprotein primarily synthesized by the thick ascending limb (TAL) of the loop of Henle [10], and is associated with improved tubular function in the general population and lower risk of AKI [11,12,13] and CKD [14]. Uromodulin knockout mice developed more aggressive renal damage in an ischemia/reperfusion injury (IRI) model than wild type mice [12,13]. Furthermore, urinary uromodulin was shown to be a biomarker of AKI [15,16]. Preclinical studies point to extending predictive or diagnostic value of uromodulin in the AKI-to-CKD transition.

A growing body of evidence indicates that overactivation of the complement system participates in the pathogenesis of AKI resulting from ischemia/reperfusion or toxins, and induces a downstream inflammatory cascade in the injured kidney [17]. The deposition of C3 [17] and membrane attack complex (MAC) [18] around tubular cells is recognized as evidence of complement activation. The complement alternative pathway (AP) was shown to trigger most of the complement activation in AKI, and selective inhibitors of AP protected the tubular cells [19,20]. Complement factor H (CFH), as a major regulatory component of AP, is reported to decrease during AKI [14,19]. Nevertheless, the detailed protection mechanism of CFH in kidney damage needs to be explored to accelerate relevant drug development.

Previously, uromodulin was confirmed to bind with CFH in vitro and served as an enhancer for CFH to help complement factor I (CFI) cleave C3b to iC3b [21,22]. We also reported that the binding capacity between uromodulin and CFH is influenced by sialic acids on uromodulin, local pH and sodium concentration, and subsequently adjusted the activation of AP [23]. In the progression of AKI to CKD, it is still unknown whether the interaction occurs in vivo and contributes to kidney defense.

Thus, we developed a rat model of AKI-to-CKD transition by a four-time injection with low-dose cisplatin to model the renal reactions during multiple chemotherapy treatments in cancer patients. Based on the model, the changes in uromodulin and CFH and their interaction in CP-induced AKI-to-CKD transition may be revealed.

## 2. Results

### 2.1. Setting Up a Rat Model of AKI-to-CKD Transition Induced by Repeated Injections with Low-Dose Cisplatin

We established a rat model of AKI-to-CKD transition using low-dose cisplatin (3.2 mg/kg) administration once every two weeks for a total of four times (Figure 1a). Rats were randomly sacrificed 4 and 14 days after each cisplatin treatment to observe kidney pathological development during the acute phase and repair stage, respectively. Blood was collected 4, 7 and 14 days after each cisplatin injection and bio traits including SCR and BUN were measured. Urinary kidney injury biomarkers, such as KIM-1 and NGAL, were detected 4 and 14 days after each cisplatin treatment.

The levels of SCR (Figure 1b) and BUN (Figure 1c) showed a significant increase on the 4th day after each exposure to cisplatin and then decreased to lower levels until the next cisplatin injection. Remarkably, the four peaks of the SCR and BUN curves showed an upward trend. After the first bolus of cisplatin, SCR returned to normal in 7 days. However, after the next three cisplatin injections, SCR and BUN levels struggled to reach normal levels within 14 days. Even 14 days after the third dose of CP, the levels of SCR and BUN remained at relatively high values, close to their first peaks. The increased peaks and delayed recovery period of SCR and BUN indicated a decreased repair ability and accumulated renal damage caused by multiple cisplatin dosage regimens.

The urine levels of KIM-1 (Figure 1d) and NGAL (Figure 1e) normalized by creatine progressively increased on each 4th day after cisplatin treatment compared with the baseline data. Repeated episodes of cisplatin injection had the same effect on the rising range of UKIM1/UCRE at each AKI stage, but presented a slower rate of decline during each chronic phase of the disease. However, with the development of CKD, there was a decline in the level of UNGAL/UCRE on each 4th day after cisplatin injection.

Kidney histology assessment of PAS and Sirius red staining (Figure 1f) revealed the transition from initial acute tubular injury to chronic interstitial fibrosis. After cisplatin re-exposure, the lesion area of renal tubular cells kept enlarging and fibrous tissue deposition continued to accumulate (Figure 1g,h).

### 2.2. Uromodulin as a Biomarker of Kidney Injury Episodes during AKI-to-CKD Transition

To investigate the effects of repeated cisplatin administration on renal uromodulin expression, protein levels and mRNA levels were tested in both renal tissue and urine samples, which were collected on the 4th and 14th days after each drug delivery. As shown in Figure 2a, after the first dose of 3.2 mg/kg cisplatin, the relative gene expression of uromodulin was diminished to 10% of the baseline (*p* < 0.01) by day 4 and recovered to 40% (*p* < 0.01) by day 14. The UMOD mRNA level decreased again at the acute phase (*p* < 0.05) after the second cisplatin injection. Unlike after the first dose, the uromodulin expression level could not rise again and remained at a low level. Subsequently, UMOD gene expression was sustained at a low level for the rest of the two cisplatin doses (*p* < 0.05). The changes in uromodulin protein level (Figure 2c,d) revealed by Western blot showed a similar pattern with mRNA level, whereas the urinary uromodulin/creatinine ratio (Figure 2b) seemed to show the opposite. Although the basal urine excretion level of uromodulin was considerably abundant in the normal rat, the four consecutive cisplatin injections all induced a further increase at day four even in the already-CKD rat. Two weeks after each cisplatin exposure, the urinary uromodulin/creatinine ratio fell to a level slightly higher than baseline. Compared with urinary KIM-1/creatine and NGAL/creatinine, urinary uromodulin/creatine might be a more suitable biomarker to monitor acute nephrotoxicity on a CKD background. Immunohistochemistry staining showed the dynamic changes of uromodulin distribution in cisplatin re-exposed rat kidneys (figure in below). In the normal kidney, strong uromodulin expression was observed in the TAL of the loop of Henle and the distal convoluted tubule. Acute nephrotoxicity sharply decreased the positive staining of uromodulin and made it shed into the lumen. After repeated cisplatin administration, positive staining intensity did not recover to previous levels (figure in below).

### 2.3. Transcriptomic Analysis of the Kidneys in the Acute Phase after the First Cisplatin Injection

Complex and diverse signal pathways have been reported to be involved in the mechanisms of CP-induced AKI, including apoptosis, cell proliferation and differentiation, inflammation and stress responses [24]. In the current study, we conducted genome-wide transcriptomic sequencing of normal kidneys and CP-treated kidneys on day 4 after the first cisplatin dose. The log10 counts per million (CPM) values were used to quantify mRNA expression. We analyzed the profiles of DEGs between the two groups. The top 20 KEGG terms of upregulated genes with an adjusted *p*-value < 0.05 were diagramed in Figure 3a. “Complement and coagulation cascades” was one of the most enriched KEGG pathways among these DEGs. Among the three complement activation pathways (Figure 3b), AP ranked first by further enrichment analysis. As diagramed in the heatmap (Figure 3c), the critical components of AP, including CFH, CFI, CFD and CFB were upregulated in the acute phase of cisplatin toxicity. Based on the previously reported interaction between uromodulin and CFH in vitro [21,22,23], we further explored the dynamic change of CFH and CFH–uromodulin interaction in the AKI-to-CKD rat model.

### 2.4. Reduced Renal CFH Is Associated with the Progression of CKD

To determine the effect of repeated cisplatin treatment on CFH, we further observed the changes of CFH in serum, kidney and liver. Repeated cisplatin hepatotoxicity was associated with downregulation of CFH expression in rat livers (Figure 4a). Specifically, the hepatic mRNA level of CFH was significantly reduced after the first two doses of cisplatin and was kept at half of the baseline level during the following two doses of cisplatin. Consequently, there was a decrease in serum CFH following each cisplatin administration (Figure 4b). Relative gene expression of renal CFH presented a time-dependent increase in the transition of AKI to CKD (Figure 4c). Remarkably, the latter two episodes of cisplatin attack significantly elevated the mRNA level of CFH in the kidneys. To investigate renal CFH, Western blots (Figure 2c,e,f) and immunohistochemistry (Figure 5a,b) were performed in kidney tissues at the nine time points mentioned above. The sustained decline in renal CFH protein level was negatively associated with the accumulation of α-SMA, a vital fibrosis marker. These results suggested a correlation between renal CFH protein reduction and AKI-to-CKD progression induced by repeated cisplatin injections.

### 2.5. The Interaction between Uromodulin and CFH May Be Decreased with the Development of AKI to CKD

Previously, we confirmed the interaction between uromodulin and CFH in vitro and how sialic acids on uromodulin, local pH and sodium concentration affected this interaction [22,23]. We performed a co-IP assay to further test whether uromodulin and CFH interacted with each other in vivo. Lysates of normal rat kidneys were prepared by weak lysis buffer. After the co-IP process using an anti-uromodulin monoclonal antibody and IgG as a negative control, CFH was identified at the position of approximately 180 kDa (Figure 6a). The smaller bands at approximately 150 kDa were verified as IgG. To confirm complement activation in kidneys, we found that deposition of C3 along the renal tubular basement membrane was progressively elevated at day 4 after the first cisplatin injection and remained increased in a time-dependent manner in the chronic phase, as measured by immunohistochemistry (Figure 6b,c). These findings suggested that complement activation was associated with a decrease in uromodulin and CFH during AKI-to-CKD progression.

## 3. Discussion

In this study, we employed a reproducible model of AKI progressing to CKD induced by multiple repeated injections of low-dose cisplatin. We observed a dynamic decrease in uromodulin and CFH protein levels, which were associated with the development of renal fibrosis and C3 deposition. We further detected binding between uromodulin and CFH in normal rat kidneys. Given the above, we propose that the decline in their interplay correlates with complement activation and may participate in the chronic fibrosis process.

Much effort in nephrology has concentrated on exploring emerging biomarkers for AKI and CKD or extended clinical applications of existing biomarkers [15]. Over the past few years, KIM-1 and NGAL have been recognized as early biomarkers of AKI which are even more sensitive than SCR at a very early phase [25,26]. Urinary uromodulin has been reported as a potential biomarker for AKI [27] and tubular dysfunction [28]. In the present study, we detected a time course of changes in renal protein and mRNA levels and urinary concentration of uromodulin during the AKI-to-CKD transition. The expression of uromodulin in kidneys was drastically reduced in the acute phase of cisplatin toxicity compared with normal controls. Meanwhile, a significant amount of uromodulin was shed from the apical membrane of renal tubular epithelial cells into the lumen and then excreted into urine [29] as demonstrated by immunohistochemistry. Taken together, the protein content of uromodulin in the kidney dropped further and the urinary concentration was increased in the early damage stage. Although protein and mRNA levels of uromodulin picked up during the chronic repair process, recurrent cisplatin exposures delayed the recovery of uromodulin. Thus, urine concentration of uromodulin was characterized by soaring in the early stage of tubular damage and then falling back to baseline in the repair phase induced by repeated AKI episodes regardless of CKD development. By contrast, elevated urine concentrations of KIM-1 and NGAL, noted for their sensitivity in detecting early renal injury [26], appeared in both the acute and chronic phase of kidney injury and could not discriminate the newly occurred damage based on CKD progression. Therefore, the clinical significance of urinary uromodulin/creatine maybe extended to identify acute tubular injury among CKD patients, especially patients using multiple doses of cisplatin. Further diagnostic validation and efficacy evaluation need to be performed in cross-sectional or cohort studies.

Thus far, complement activation within the kidney is considered as a secondary event in AKI, and clinical studies have revealed that activation of the AP is central to the pathogenesis of AKI [17,30]. In patients with congenital defects in CFH, the kidney is the most common target of AP activation [31]. Conversely, tubular epithelial cells cultured with CFH showed a significant regulation effect on complement activation on the apical and the basal surfaces of the cells [32]. As we detected, the protein levels of CFH—which are mainly derived from plasma in normal rat kidneys—were comparatively low, and AKI further reduced the inadequate concentration. Transcriptome sequencing suggested that renal production of complement proteins extensively increased, which was reported to promote local complement activation and injury [17]. It was confusing that the dynamic trends of CFH in mRNA levels were contrary to protein levels. A possible explanation for these observations could be that the local CFH mRNA accumulation failed to translate to sufficient CFH protein to fill the gap. That is, the local mRNA level of CFH was not the decisive factor for renal CFH protein content. Spatial and single-cell transcriptomics may help elucidate the expression profile of CFH and explain the contradictory trends during AKI-to-CKD transition.

Interactions between uromodulin and CFH have been reported, especially in vitro [21,22,23]. Indeed, uromodulin was confirmed to bind strongly with several complement components involving three pathways, such as c1q [33,34] and collectin-11 [35]. However, uromodulin is mainly abundant in urine while complement components are mainly abundant in blood, limiting the studies on the interactions between them in vivo. In our study, we inputted excessive rat kidney tissue to prepare the protein lysates for co-IP assays and detected CFH by Western blotting in the elution product incubated with anti-uromodulin antibody. With the decrease of uromodulin and CFH in kidneys during the AKI-to-CKD transition, we speculated that the interaction between them declined simultaneously. Meanwhile, the changes in C3 deposition along the renal tubular basement membrane proved the complement activation [36] associated with the loss of uromodulin and CFH. Whether the interaction decline was responsible for the complement activation remains to be further investigated. Future studies may focus on the specific binding sites, sequences or glycosylation and clinical application for AKI-to-CKD prevention. Owing to the central position of AP in the pathogenesis of AKI [17], the interaction between uromodulin and AP is of profound translational value.

## 4. Materials and Methods

### 4.1. Animals

Male Sprague Dawley (SD) rats (6–7 weeks old, body weight 180–200 g) were obtained from Vital River Laboratory Animal Technology (Beijing, China). All animals were maintained in animal facilities under specific-pathogen-free (SPF) conditions. They had free access to standard rat chow and water. They lived in a room with constant temperature and humidity with a 12:12 h light–dark cycle. Animal experiments were approved by the Institutional Animal Care and Use Committee at Peking University First Hospital (Approval Number J202054).

### 4.2. Experimental Protocol

After adaptive feeding for a week, healthy SD rats were randomly divided into the normal saline (NS) group (n = 6) or the CP group (n = 42) (Figure 1a). Cisplatin (P4394, Sigma-Aldrich, Germany) was dissolved in sterile 0.9% NaCl at a concentration of 1 mg/mL. The CP group was administrated cisplatin intraperitoneally (i.p.) after 12 h starvation. The single dose of cisplatin was 3.2 mg/kg of body weight. The same dosage injection of cisplatin was repeated every 14 days for a total of 4 times. The NS rats were administered the same volume of sterile 0.9% NaCl (Figure 1a). At the 4th, 7th and 14th day after each injection, 12 h urine and blood sample were collected and baseline samples were used as controls. Before the first injection at day 0, we randomly sacrificed 6 rats. Then, rats in the CP group (6 per day) were sacrificed on the 4th, 14th, 46th and 56th day after the first cisplatin treatment, respectively; and 3 per day on 18th, 28th, 32th and 42th day (Figure 1a). The kidneys and livers were dissected immediately following sacrifice, then stored at −80 °C or preserved in formalin.

### 4.3. Blood and Urine Examination

Serum creatinine (SCR) and urine creatinine were measured by Creatinine (Cr) Assay kit (C011-2-1, Nanjing Jiancheng Bio., Nanjing, China). Blood urea nitrogen (BUN) was tested by a Urea Assay kit (DIUR-100, BioAssay Systems, Hayward, CA, USA). The concentration of serum CFH was measured using a Rat CFH ELISA kit (SEA635Ra, Cloud-Clone Corp., Wuhan, China) according to the manufacturer’s instructions. Cloud-Clone Corp. also provided an ELISA kit to detect the concentration of urine uromodulin (SEB918Ra) and neutrophil gelatinase-associated lipocalin-2 (NGAL) (SEB388Ra). Urinary kidney injury molecule 1 (KIM-1) was tested by Rat KIM-1 Quantikine ELISA kit (RKM100, R&D Systems, Minneapolis, MN, USA). The level of urinary uromodulin, KIM-1, NGAL and CFH were normalized to urine creatinine.

### 4.4. Renal Histology

Kidney tissue sections were fixed with 10% buffered formalin followed by paraffin embedding. Periodic acid–Schiff (PAS) (BA4114, Baso, Zhuhai, China) stain was used to evaluate tubular atrophy and Sirius red (G1472, Solarbio, Beijing, China) stain to show collagen fibers. The calculation was determined by 10× magnification in 10 fields for each section and performed in a blinded fashion. Tubular atrophy is defined as foamy degeneration, tubular necrosis, detachment, cast formation and dilation, and was semiquantitatively assessed with a score of 0 to 4 according to the lesion area percentage (0 = no lesion, 1 = <25%, 2 = 25–50%, 3 = 51–75%, 4 = 76–100%). As for fibrosis semiquantification, a ratio of red stained area to the entire field (glomeruli, tubule lumina and blood vessels, if any, excluded) was assessed by Image-Pro Plus 6.0 and expressed as a percentage of fibrotic area.

### 4.5. RNA Sequencing

Before (n = 3) and 4 days after the first cisplatin injection (n = 3), kidney samples of 3 randomly selected rats were extracted for RNA sequencing. Total RNA was extracted using the mirVana miRNA Isolation Kit (AM1561, Invitrogen, Waltham, MA, USA) following the manufacturer’s protocol. RNA integrity was evaluated using the Agilent 2100 Bioanalyzer (Agilent Technologies, Santa Clara, CA, USA). The samples with RNA Integrity Number (RIN) ≥ 7 were used for subsequent analysis. The libraries were constructed using a TruSeq Standard mRNA LTSample Prep Kit (RS-122-2101, Illumina, Santa Clara, CA, USA), then sequenced on an Illumina sequencing platform (HiSeqTM 2500 or Illumina HiSeq × Ten) and 150 bp paired-end reads were generated. Transcriptome sequencing and analysis were conducted by OE Biotech Co., Ltd. (Shanghai, China). Raw data were processed using Trimmomatic. Reads containing poly-N and low-quality reads were removed to obtain clean reads. Clean reads were then mapped to the reference genome using HISAT2.

Fragment per kilobase of exon model per million mapped fragments (FPKM) and read count values of each transcript were calculated using bowtie and eXpress. Differentially expressed genes (DEGs) were identified by the DESeq functions estimateSizeFactors and nbinom test. *p* value < 0.05 and fold change > 2 or fold change < 0.5 were set as the threshold for significantly differential expression. Gene Ontology (GO) enrichment and Kyoto Encyclopedia of Genes and Genomes (KEGG) pathway enrichment analysis of DEGs were respectively performed using R based on the hypergeometric distribution.

### 4.6. Real-Time PCR (RT-PCR)

Kidney and liver tissues were collected in RNase-free tubes, and total RNA was isolated using TRIzol reagent (15596026, Thermo Fisher, Waltham, MA, USA) following the instructions of the manufacturer. Photometric measurements were used to check the RNA concentration and purity. For cDNA synthesis, 1 μg of total RNA was used for reverse transcription with HiScript III All-in-one RT Supermix Perfect for qPCR (R333, Vazyme, Nanjing, China). The sequences of the primers used are listed in Table 1. RT-PCR was to detect the ct values of the genes with PowerUp SYBR Green Master Mix (A25742, Thermo Fisher, USA). Comparative gene expression was calculated by the 2^−∆∆Ct^ method.

### 4.7. Antibodies

The antibodies used for Western blot and immunohistochemistry (IHC) included rabbit monoclonal anti-uromodulin antibody (ab207170, Abcam, Cambridge, UK), mouse monoclonal anti-Factor H antibody (ab118820, Abcam, UK), mouse monoclonal anti-alpha-smooth muscle actin (α-SMA) (ab7817, Abcam, UK), mouse anti-uromodulin antibody (sc-271022, Santa Cruz Bio., Dallas, TX, USA), mouse monoclonal anti-GAPDH antibody (ab8245, Abcam, UK), and rabbit monoclonal anti-C3 antibody (ab200999, Abcam, UK). Secondary antibodies for immunoblotting were horseradish peroxidase (HRP)-labeled goat anti-mouse IgG (H + L) (ZB-2305, ZSGB-BIO, Beijing, China) and mouse anti-rabbit IgG-HRP (sc-2357, Santa Cruz Bio., Dallas, TX, USA).

### 4.8. Western Bolts and Co-Immunoprecipitation (Co-IP)

Total protein from kidney samples of SD rats was extracted on ice with Cell Lysis Buffer for Western Blot and IP (P0013, Beyotime, Nantong, China) supplemented with protease and phosphatase inhibitor cocktail (P1051, Beyotime, China) following standard protocols. We used the Pierce Classic Magnetic IP/Co-IP kit (88804, Thermo Fisher, USA) to perform the co-IP experiment. Protein concentration was measured with a Pierce BCA Protein Assay kit (23227, Thermo Fisher, Rockford, IL, USA). The proteins were denatured for reduced gels except when we aimed to detect CFH. Protein samples were separated in sodium dodecyl sulfate–polyacrylamide gels and then were electrically transferred onto polyvinylidene difluoride membranes. The membranes were blocked in 5% fat-free milk dissolved in Tris-buffer saline with 0.1% Tween 20 (TBST) for 1 h at room temperature and incubated with primary antibodies at 4 °C overnight. After washing three times with TBST, the membranes were incubated with HRP-conjugated secondary antibodies for 1 h at room temperature. After three washes with TBST, the membranes were incubated in Immobilon^®^ ECL Ultra Western HRP Substrate (WBULS0500, Millipore, Billerica, MA, USA), and images were captured by an ImageQuant LAS 4000 mini system (GE, Healthcare, Pittsburgh, PA, USA). The relative intensity of the protein bands was quantified by digital densitometry using ImageJ 1.52i software (Media Cybernetics, Rockville, MD, USA). The level of GAPDH expression was used as an internal standard.

### 4.9. Immunohistochemistry (IHC)

IHC staining of the kidney was performed on paraffin sections. After antigen retrieval, sections were incubated overnight with primary antibodies at 4 °C. Having washed with PBS three times, the slides were exposed to HRP-labeled secondary antibodies (PV-6000, ZSGB-BIO, Beijing, China) and DAB substrate (ZLI-9018, ZSGB-BIO, Beijing, China) to visualize antigens. Finally, the slides were counterstained with hematoxylin and mounted after dehydration. For quantification, 10 fields (40×) were randomly selected from each tissue section and the staining was analyzed using Image-Pro Plus 6.0 in a blinded manner.

### 4.10. Statistical Analysis

Statistical analysis was performed using SPSS 25.0 statistical software (SPSS, Chicago, IL, USA). Variables distributed normally were shown as mean ± SEM and compared using a *t*-test, while non-normally distributed or nonparametric variables as median and interquartile range and compared between groups using the Mann–Whitney U-test. All *p*-values were two-tailed, and *p* < 0.05 was considered significant. GraphPad Prism 8.0 was used to visualize the vector diagrams (GraphPad, San Diego, CA, USA).

## 5. Conclusions

In the current study, we established a rat model of AKI-to-CKD transition induced by repeated cisplatin treatments to mimic chemotherapy routines of cancer patients: reversible AKI and recurrent AKI on the background of CKD progression. We elaborately demonstrated the dynamic changes in the expression and excretion of uromodulin along with renal dysfunction and fibrosis deposition. Compared with SCR, KIM-1 and NGAL, urine uromodulin exhibited superior ability in discriminating emerging acute tubular injury on CKD status. We also showed the binding of uromodulin and CFH in vivo for the first time. Along with the CFH reduction and C3 deposition during the AKI-to-CKD transition, the underlying mechanism of uromodulin–CFH interaction and its participation in disease development opened up a new prospective research avenue.

## Figures and Tables

**Figure 1 ijms-24-06636-f001:**
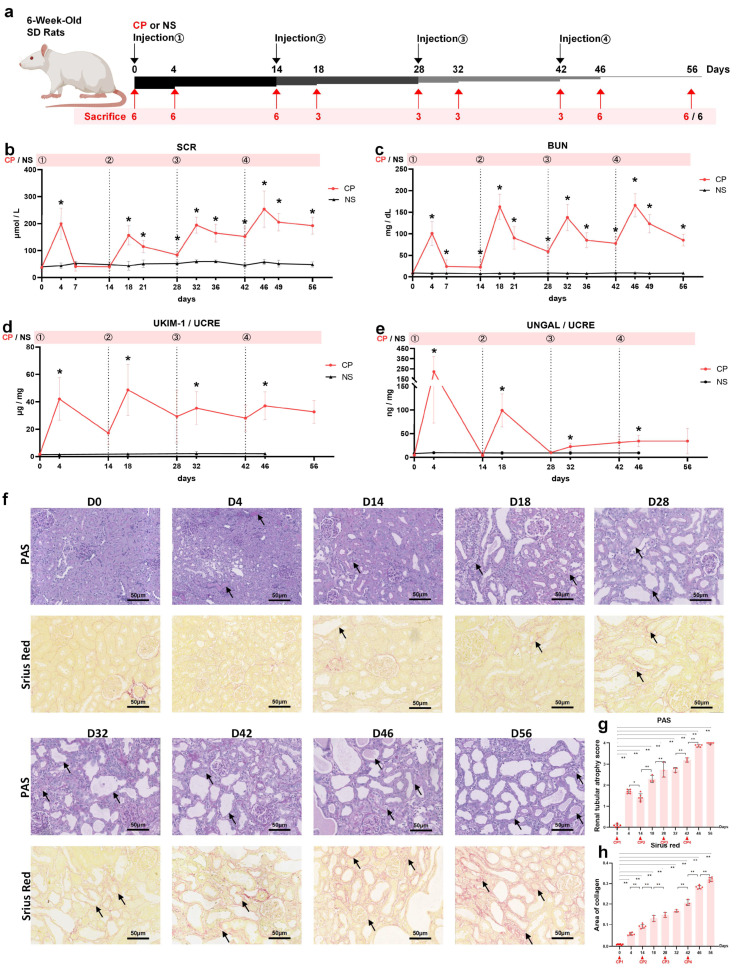
The four-time intraperitoneal injection of low-dose cisplatin (CP) induced AKI-to-CKD transition in rats. (**a**) Study design overview. Rats (6 weeks old) were injected with 3.2 mg/kg cisplatin every 14 days as the CP group and 6 of them were randomly sacrificed at days 0, 4, 14, 46 and 56; and 3 at days 18, 28, 32 and 42. The normal control (NC) group was injected with the same volume of sterile 0.9% NaCl and was maintained for 56 days. Changes in (**b**) SCR, (**c**) BUN, (**d**) urinary KIM-1/UCRE and (**e**) urinary NGAL/UCRE levels in rats during the 56 days (n = 6). (**f**) Representative photomicrographs of PAS and Sirius red. Arrows point to the renal tubular injury in PAS-stained images and renal fibrosis in Sirius-red-stained images. (Original magnification 20×; Scale bar = 50 μm). Quantification of (**g**) PAS and (**h**) Sirius red. Data are presented as means ± SD. * *p* < 0.05, ** *p* < 0.01 between the two groups.

**Figure 2 ijms-24-06636-f002:**
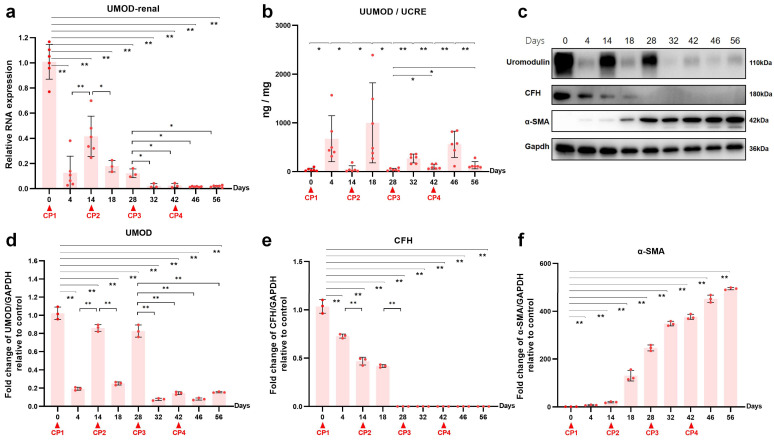
Uromodulin expression during cisplatin (CP)-induced AKI-to-CKD transition in rats. (**a**) Relative expression of uromodulin in kidney after normalization with GAPDH. (**b**) Urinary uromodulin level normalized by UCRE (n = 6). (**c**) Representative Western blots of uromodulin, CFH and α-SMA. The relative protein expression of (**d**) uromodulin, (**e**) CFH and (**f**) α-SMA to GAPDH at different time points (n = 3). Data are presented as means ± SD. * *p* < 0.05 and ** *p* < 0.01 between the two groups.

**Figure 3 ijms-24-06636-f003:**
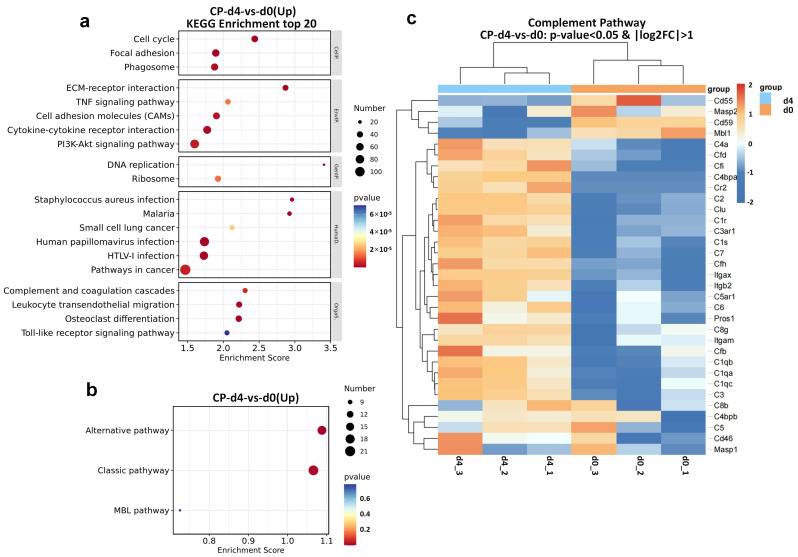
Transcriptomic analysis of kidneys from SD rats before and 4 days after the first cisplatin (CP) injection. (**a**) The top 20 enriched KEGG pathways of significant DEGs (total). (**b**) The enrichment of three complement activation pathways of significant DEGs (up). (**c**) Heatmap showing the complement-pathway-related DEGs differentially expressed across the two groups, n = 3. KEGG, Kyoto Encyclopedia of Genes and Genomes; DEGs, differentially expressed genes; Clu, clusterin; Itgax, integrin subunit alpha X; Itgb2, integrin subunit beta 2; Pros1, protein S (alpha).

**Figure 4 ijms-24-06636-f004:**
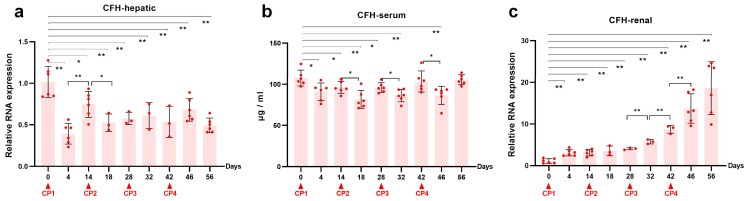
CFH expression during cisplatin (CP)-induced AKI-to-CKD transition in rats. Relative expression of CFH mRNA in (**a**) liver and (**c**) kidney after normalization with GAPDH. (**b**) Quantification of serum CFH level by ELISA with a 500-fold dilution of serum. Data are presented as means ± SD. * *p* < 0.05 and ** *p* < 0.01 between the two groups.

**Figure 5 ijms-24-06636-f005:**
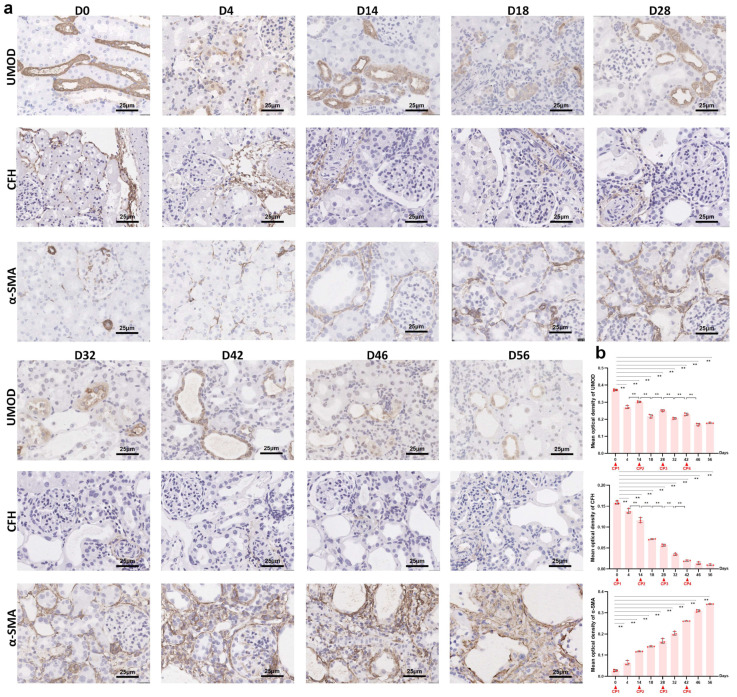
Immunohistochemical staining of uromodulin, CFH and α-SMA in kidneys on a series of days. (**a**) Representative micrographs (40×, bar = 25 μm) and (**b**) quantification of positive staining on kidney sections on different days as indicated (n = 3). ** *p* < 0.01 between the two groups. CP, cisplatin.

**Figure 6 ijms-24-06636-f006:**
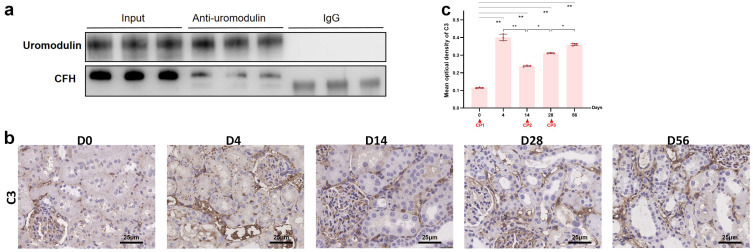
Interaction between uromodulin and CFH in rat kidneys and complement activation during AKI-to-CKD transition. (**a**) Total protein lysates of normal rat kidneys were immunoprecipitated using an antibody against uromodulin and immunoblotted against CFH. (**b**) Representative micrographs (40×, bar = 25 μm) and (**c**) quantification of positive-stained C3 on kidney sections on different days as indicated (n = 3). * *p* < 0.05, ** *p* < 0.01 between the two groups. CP, cisplatin.

**Table 1 ijms-24-06636-t001:** Primer sequences used for RT-PCR.

Gene	Primer Sequences	
UMOD	Forward	5′-TGCTGGAAACTATGACCTAG-3′
	Reverse	5′-GATGGGACCCAAGTTCAGGA-3′
CFH	Forward	5′-CCCTAATTTCCCAACGTGTG-3′
	Reverse	5′-TCATATTCCACCACGTCACC-3′
GAPDH	Forward	5′-TGGCCAAGGTCATCCATGA-3′
	Reverse	5′-GCAGTGATGGCATGGACTGT-3′

## Data Availability

The data presented in this study are available in the article.

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
