# Peer review of "The Reduction of Uromodulin, Complement Factor H, and Their Interaction Is Associated with Acute Kidney Injury to Chronic Kidney Disease Transition in a Four-Time Cisplatin-Injected Rat Model"

_ijms, 2023, doi:10.3390/ijms24076636_

Round 1

Reviewer 1 Report

This is a nice paper. However, I have some comments. The findings from this paper are excellent and worthy to review. This manuscript contained some questions described below. I think this paper is interesting, this review contributes to future's clinical medicine largely. I have some questions from a point of view of clinical medicine. This paper is about one mechanism of renal failure progression that could be explained by biomarker trends. Why did you focus on NGAL and KIM1 as biomarkers? Secondly, I think that inflammation is the key word for the transition to CKD, and I think that circulatory disturbance is the trigger for inflammation. In particular, I think evaluation of urine L-FABP, a biomarker of tubular ischemia, is necessary, but would appreciate your comments. In addition, NAG and β2microglobulin are also frequently used as markers of tubular damage in clinical practice, but how are these biomarkers evaluated? Please let me know. If we use drugs such as SGLT2 inhibitors and ARNI, which are speculated to have renoprotective effects, in this tubular injury model, we may be able to see such data trends.

Reviewer 2 Report

This is a very interesting manuscript by Xing et.al, investigated the “The Reduction of Uromodulin, Complement Factor H and their Interaction Associated with Acute Kidney Injury to Chronic Kidney Disease Transition in a Four-time Cisplatin Injected Rat Model”. The authors establishes uromodulin interacts with FH in kideny and decreases with cisplatin treatment. Overall the methodologies and assays chosen are sound and experiments appear to have been conducted carefully, and the conclusions made are well-founded with the presented results. However, some points can be raised, as indicated below.

Line 37: Should be “major” excretory pathway

Line 86-87: what is the rationale behind in selecting Cisplatin dose as 3.2mg/kg for 4 times and any previous literature?

Figure 1b and 1c: what are the additional timepoints (after d18, d32 and day 46 which showed significant represented as *

Figure 1d and 1e: concentration of UKIM-1/UCRE and UNGAL/UCRE in normal saline group is missing.

Line 119: arrows showing lesion area and acute tubular injury in figure is missing

Line 124: should be protein “levels”

Line 124: Should be “tested in both renal tissue and”

Line 126: Should be first dose of cisplatin

Line 129: Should be “unlike after first dose”

Line 131: Should be cisplatin doses

Line 132: Should be “Western blot”

Line 132: Is it reducing on nonreducing gel?

Line 143: Remove “positive”

Line 145: Remove “strength”

Line 148: Remove “evolution of”

Line 150:  Should be “Western”

Line 150: Figure 1c: are all these Western blot images are from same PVDF membrane?

Line 151: Figure 1c: D0, d4 and d18 Gapdh protein levels are different from rest of the samples.

Line 151: Figure 1c: why no uromodulin was not seen in d32 sample?. Should be “kDa” for kilodalton

Lone 159: Should be CP “treated mice”

Line 160: Should be cisplatin “dose”

Line 164: Should be “among” the three

Line 165: Reference for ulterior enrichment analysis

Line 173: What are Clu, Itgax, Itgb2, Pros 1, ?

Line 180: Should be “two “doses”

Line 181: Should be “two “doses”

Line 186: Remove “protein contents of”

Line 186: Should be “Western blot”

Line 188: Renal FH mRNA were increasing in fig.4c, while FH protein levels were decreased as shown in fig.1C, 1e. How do you justify?

Line 193: Remove “evolution”

Line 194: Should be (a) Liver and (c) kidney

Line 195: Should be (b) quantification

Line 195: CFH concentration (µg/ml) in serum represented in figure 4b shows maximum 80-100 µg/ml but rat FH concentration has been reported to be 238 + 21ug/ml by Demberg T et al., J. Immunol. 56:149-160, 2002.

Line 208: Should be “uromodulin and”

Line 210: Why IgG bands are appeared when detected with anti-FH and is it reducing or nonreducing gel?

Line 219: Why uromodulin in input and anti-uromodulin samples looks same?

Line 307: any significant difference in kidney and liver organ weights?

Line 313: CFH ELISA kit (SEA635Ra) range is 15-1000ng/ml but in results CFH levels are shown as >80µg/ml.

Line 323: Should be each “section”

Line 356: Should be primers “used were listed in Table 1 “

Line 356: Remove “were tested with excellent performance”

Line 357: Remove “applied”

Line 363: Should be “Western”

Line 364: does anti-FH (ab118820) cross reacts with rat FH?

Line 413: “decrease of C3 deposition” statement contradicts the claim in line 211-214.
